# Salvage versus Primary Liver Transplantation for Hepatocellular Carcinoma: A Twenty-Year Experience Meta-Analysis

**DOI:** 10.3390/cancers14143465

**Published:** 2022-07-16

**Authors:** Gian Piero Guerrini, Giuseppe Esposito, Tiziana Olivieri, Paolo Magistri, Roberto Ballarin, Stefano Di Sandro, Fabrizio Di Benedetto

**Affiliations:** Hepato-Pancreato-Biliary Surgery and Liver Transplantation Unit, Policlinico Modena Hospital, Azienda Ospedaliero Universitaria di Modena, Via del Pozzo 71, 41125 Modena, Italy; giuseppe.esposito@unimore.it (G.E.); olivieri.tiziana@aou.mo.it (T.O.); paolo.magistri@unimore.it (P.M.); ballarin.roberto@aou.mo.it (R.B.); sdisandro@unimore.it (S.D.S.); fabrizio.dibenedetto@unimore.it (F.D.B.)

**Keywords:** HCC, salvage liver transplantation, rescue liver transplantation, liver transplantation, liver resection

## Abstract

**Simple Summary:**

Primary liver transplantation (PLT) for HCC represents the ideal treatment. However, since organ shortage increases the risk of drop-out from the waiting list for tumor progression, a new surgical strategy has been developed: Salvage Liver Transplantation (SLT) can be offered as an additional curative strategy for HCC recurrence after liver resection. The aim of this updated meta-analysis is to compare surgical and long-term outcomes of SLT versus PLT for HCC. The findings of our analysis reveal that SLT offers comparable surgical outcomes but slightly poorer oncological long-term outcomes with respect to PLT.

**Abstract:**

(1) Background: Primary liver transplantation (PLT) for HCC represents the ideal treatment. However, since organ shortage increases the risk of drop-out from the waiting list for tumor progression, a new surgical strategy has been developed: Salvage Liver Transplantation (SLT) can be offered as an additional curative strategy for HCC recurrence after liver resection. The aim of this updated meta-analysis is to compare surgical and long-term outcomes of SLT versus PLT for HCC. (2) Materials and Methods: A systematic review and meta-analysis was conducted using the published papers comparing SLT and PLT up to January 2022. (3) Results: 25 studies describing 11,275 patients met the inclusion criteria. The meta-analysis revealed no statistical difference in intraoperative blood loss, overall vascular complications, retransplantation rate, and hospital stay in the SLT group compared with the PLT group. However, the SLT group showed a slightly significant lower 5-year OS rate and 5-year disease-free survival rate. (4) Conclusion: meta-analysis advocates the relative safety and feasibility of both Salvage LT and Primary LT strategies. Specifically, SLT seems to have comparable surgical outcomes but slightly poorer long-term survival than PLT.

## 1. Introduction

Hepatocellular carcinoma (HCC) is a major contributor to the world’s cancer burden and is currently the third leading cause of cancer-related death, with incidences increasing continuously in recent years [1]. Locoregional treatments (mainly radiofrequency ablation and transarterial chemoembolization), liver resection (LR), and liver transplantation (LT) are well-defined and widely accepted treatments for hepatocellular carcinoma [2,3,4]. However, the best therapy for HCC is still an open and controversial oncological challenge. LT is considered the gold standard therapy for early HCC within liver cirrhosis since it radically removes the cancer and any dysplastic foci and it treats liver disease-related complications (e.g., portal hypertension) [5,6,7]. The oncological benefits of LT for HCC in terms of 5-year overall survival (OS) and 5-year disease-free survival (DFS) are well-documented: 75% and 90%, respectively [8]. However, organ shortage and the risk of drop-out from the waiting list for tumor progression and deterioration of liver function represent the main limitations for LT [9]. Nowadays, liver surgery for HCC has been demonstrated to be feasible and safe with very low postoperative morbidity and almost zero perioperative mortality [10,11]. Studies on minimally invasive liver surgery have also strongly confirmed these findings [12]. Therefore, primary LR for early HCC with preserved liver function and mild portal hypertension is considered the first-choice treatment [13,14]. Nevertheless, most published data showed a 5-year survival rate and a 5-year DSF after LR for HCC due to cancer relapses of 60% and 30%, respectively [15]. Salvage Liver Transplantation (SLT) is an alternative and promising curative strategy for HCC recurrence or deterioration of liver function after primary liver resection [16]. Moreover, some authors recently described “de principe” Salvage LT (pre-emptive transplantation before tumor recurrence) for a subgroup of patients who present poor histological features and aggressive biological tumor behavior on the final pathology of the resected specimen [17]. Previous studies comparing SLT with primary liver transplantation (PLT) have reported conflicting results in terms of surgical complication and risk of HCC recurrence [18,19,20,21]. However, with the advancement of surgical techniques, recent papers have shown SLT to be an effective and feasible treatment for patients with HCC recurrence after primary liver resection with a good long-term survival rate [22]. The purpose of this meta-analysis is to investigate the technical, postoperative, oncological and survival outcomes of PLT compared with SLT.

## 2. Materials and Methods

### 2.1. Study Design

Our meta-analysis was designed according to the Preferred Reporting Items for Systematic Reviews and Meta-Analyses (PRISMA) statement [23], while the authors predetermined the eligibility criteria for the study. Two investigators (E.G. and G.G.P.) independently searched the literature. All retrospective clinical studies that compared Salvage LT with Primary LT for HCC were included in the present systematic review. No prospective studies have been published so far. Case reports, reviews, letters, and animal studies were excluded. All discrepancies during the data collection, synthesis, and analysis were resolved by the consensus of two authors (E.G. and G.G.).

### 2.2. Literature Search and Data Collection

We systematically searched the literature using the PubMed, MEDLINE, and Cochrane library databases for articles published up to January 2022; querying three databases maximizes the probability of capturing articles, as recently demonstrated by Goossen et al. [24]. Our search included the words “HCC”, “salvage liver transplantation”, “rescue liver transplantation”, and “salvage liver transplantation or liver transplantation and liver resection”. The search strategy was confined to English language papers and is described in Appendix A [23] and Appendix A.

### 2.3. Quality Assessment

The quality of the included articles was estimated using the Methodological Index for Non-Randomized Studies (MINORS) [25].

### 2.4. Statistical Analysis

Meta-analysis was realized using the software Review Manager (RevMan) [Version 5.1. Copenhagen: The Nordic Cochrane Centre, The Cochrane Collaboration, 2011). Dichotomous outcomes are displayed as odds ratios (OR) with a 95% confidence interval (CI) by using the Mantel–Haenszel method and continuous variables are displayed as Mean difference (MD) with a 95% CI by utilizing the generic inverse variance method. Mean and standard deviation (SD) for continuous data, if not reported, were estimated using the method illustrated by Hozo et al. [26]. However, for continuous data provided as median and interquartile range (IQR), mean and SD were estimated by employing the method described by Luo et al. [27] and Wan et al. [28], respectively. The cut-off for statistical significance was set at *p* ≤ 0.05. Heterogeneities between the studies were evaluated using Q statistics and total variation was computed by I^2^. A random-effects model (REM) was always adopted due to the conceptual heterogeneity of clinical studies. Publication bias of the included papers is illustrated in Appendix A.

## 3. Results

### 3.1. Studies and Patient Characteristics

Our search strategy disclosed 857 publications concerning Salvage LT. Twenty-nine full papers were examined; however, five studies were not included in the analysis because they did not meet the inclusion criteria. Finally, 25 articles and a total of 11,275 patients were included in the meta-analysis; 9645 patients were offered a Primary LT for HCC, whereas 1630 underwent Salvage LT for HCC recurrence or impaired liver function after primary liver resection. No randomized trials have been published so far. The flow diagram in Figure 1 shows the search process. The baseline characteristics of the two groups are presented in Table 1 and Table 2. Technical and postoperative outcomes and oncological and survival features are tabulated in Table 3. The two groups were similar as regards etiology, HBV, and/or HCV infection rates and maximum tumor diameter pre-LT and on post-LT pathology. The number of patients in each study ranged from a minimum of 42 to up to 6975. The MINORS scale assessed a low-quality heterogeneity between studies, providing a mean score of 21.8 (SD: 0.85) and a median score of 22 (range 20–23) (Table 1).

### 3.2. Technical Outcomes

#### 3.2.1. Duration of Surgery

The mean operating time was 600.44 min in the SLT group and 547.12 min in the PLT group; sixteen articles reported this item. Operating time was shorter in the Primary LT group, and the meta-analysis showed a statistically significant difference (MD 33.30, (95% CI 17.60, 49.00) *p* < 0.0001), as shown in Figure 2.

#### 3.2.2. Intraoperative Blood Loss, Intraoperative Red Blood Cell (RBC), and Fresh Frozen Plasma (FFP) Transfusion

The meta-analysis showed no statistically significant increased intraoperative blood loss in the Salvage LT group when compared with the Primary one (MD 290.35, (95% CI −82.63, 663.32) *p* = 0.13), as shown in Figure 3. The mean intraoperative blood loss in the SLT and PLT groups was 3174.55 cc and 2342.02 cc, respectively. The mean of intraoperative RBC and FFP transfusion was 7.8 RBC units and 9 FFP units in the Salvage LT group, and 6.5 RBC units and 8 FFP units in the Primary LT group. However, our analysis revealed no statistically significant differences between the two approaches: (MD 0.92, (95% CI −0.48, 2.32) *p* = 0.07) and (MD 0.34, (95% CI −0.69, 1.36) *p* = 0.52), respectively, as shown in Figure 4 and Figure 5.

#### 3.2.3. Reoperation Rate

Reoperation rate was 16.96% (48/283) in the SLT group and 9.45% (103/1090) in the PLT group. The meta-analysis showed a statistically significant difference in the rate of reoperation between the two groups, higher in the SLT than in the PLT group (OR 2.34, (95% CI 1.53, 3.59) *p* < 0.0001), as shown in Figure 6.

#### 3.2.4. Perioperative Mortality Rate

Perioperative mortality rate was 6.31% (32/507) in the SLT group and 4.47% (100/2235) in the PLT group; slightly higher in the former group. The meta-analysis of the 18 trials showed a statistically significant difference in the rate of perioperative mortality between the two groups (OR 1.83, (95% CI 1.18, 2.84) *p* = 0.007), as shown in Figure 7.

#### 3.2.5. Retransplantation Rate

Seven studies reported the retransplantation rate. The retransplantation rate was 6.11% (8/131) in the Salvage LT group and 7.22% (70/969) in the PLT sample. However, the different rates were not statistically significant between the two treatment strategies (OR 1.07, (95% CI 0.51, 2.24) *p* = 0.86), as shown in Figure 8.

### 3.3. Postoperative Outcomes

#### 3.3.1. Postoperative Bleeding

Ten studies reported the postoperative bleeding rate. The Salvage LT group’s postoperative bleeding rate was considerably higher than the Primary LT group: 8.25% (88/1066) and 5.73% (411/7165), respectively. The difference in bleeding rates was statistically significant (OR 2.19, (95% CI 1.25, 3.81) *p* = 0.006), as shown in Figure 9.

#### 3.3.2. Intensive Care Unit Stay

The mean Intensive Care Unit (ICU) stay was 8.34 days in the SLT group and 5.44 days in the PLT group. No statistically significant mean difference was recorded (MD −0.12, (95% CI −0.88, 0.63) *p* = 0.75), although a higher mean ICU stay was displayed in the SLT group, as shown in Figure 10.

#### 3.3.3. Length of Hospitalization

The mean hospital stay was 33.01 days in the SLT group and 26.44 in the PLT group; nine articles described this variable. The meta-analysis reported that the mean hospitalization was shorter in the PLT group than in the Salvage LT group, although this imbalance was not significant (MD 0.49, (95% CI −2.13, 3.11) *p* = 0.71), as shown in Figure 11.

#### 3.3.4. Overall Vascular Complication

The rate of vascular complications was evaluated by 12 studies. The vascular complications rate was similar between SLT and PLT: 4.68% (55/1176) and 3.48% (258/7404), respectively. The meta-analysis revealed a statistically significant difference (OR 1.37, (95% CI 1.01, 1.86) *p* = 0.04), as shown in Figure 12.

#### 3.3.5. Arterial Thrombosis

A total of 34 patients developed arterial thrombosis in twelve studies. The arterial thrombosis rate in the SLT group was higher than within the PLT group: 5.56% (12/216) and 2.78% (22/790), respectively. However, a statistically significant difference in these rates was not recognized between the two approaches (OR 1.87, (95% CI 0.87, 4.03) *p* = 0.11), as shown in Figure 13.

#### 3.3.6. Biliary Complications

Thirteen papers analyzed the frequency of biliary complications (stenosis, leakage, and fistula). The biliary complication rate was significantly higher in the SLT group than the PLT group: 13.6% (162/1191) and 11.2% (838/7449), (OR 1.22, (95% CI 1.01, 1.47) *p* = 0.04), as shown in Figure 14.

#### 3.3.7. Infection and Sepsis

Ten studies retrospectively assessed overall infection and sepsis rate. Infection rate of the SLT group was slightly higher than the PLT group: 28.2% (299/1059) and 25.5% (1826/7149), respectively. Nevertheless, the meta-analysis stated that the result was not significant (OR 1.14, (95% CI 0.98, 1.32) *p* = 0.08), as shown in Figure 15.

### 3.4. Oncological and Survival Outcomes

#### 3.4.1. Overall Survival Rates

Thirteen, fifteen, and twenty studies reported the 1-year, 3-year, and 5-year overall survival (OS) rate, respectively. Our meta-analysis revealed a similar 1-year OS rate of 77.9% (1072/1375) in the SLT group and 78.5% (6801/8666) in the PLT group, although this evidence was not statistically significant (OR 0.80, (95% CI 0.62, 1.03) *p* = 0.08), as shown in Figure 16. On the other hand, the meta-analysis showed a statistically significant difference in the 3-year and 5-year OS rates between the two groups with a slightly lower OS rate in the SLT group: SLT 59.3% (837/1410) and PLT 61.9% (5508/8905) (OR 0.72, (95% CI 0.60, 0.86) *p* = 0.0002), as shown in Figure 17; and SLT 53.9% (810/1503) and PLT 56.5% (5327/9424) (OR 0.68, (95% CI 0.56, 0.82) *p* < 0.0001), as shown in Figure 18, respectively.

#### 3.4.2. HCC Recurrence Rate

Types of HCC recurrence after LT were locoregional and/or systemic. Ten studies assessed tumor recurrence rate. Disease recurrence rate was 15.4% (37/240) in the SLT group and 10.9% (98/896) in the Primary LT group. The meta-analysis showed a statistically significant difference in the rate of disease recurrence between the two groups with a lower rate in the PLT group (OR 1.93, (95% CI 1.23, 3.04) *p* = 0.004), as shown in Figure 19.

#### 3.4.3. Disease-Free Survival Rates

Twelve, fourteen, and eighteen papers retrospectively assessed the 1-year, 3-year, and 5-year disease-free survival (DFS) rates, respectively. The meta-analysis showed statistically significant differences in the DFS rate of HCC between the two groups with the same 1-year DFS rate in the SLT group and PLT group, with 71.2% (967/1358) and 71.2% (5855/8218), respectively (OR 0.66, (95% CI 0.47, 0.92) *p* = 0.01), as shown in Figure 20. However, the 3-year and 5-year DFS rates were lower in the SLT group than the PLT group: SLT 54.8% (763/1393) and PLT 57% (4821/8457) (OR 0.59, (95% CI 0.44, 0.88) *p* = 0.007), as shown in Figure 21; and SLT 49.1% (721/1468) and PLT 51.3% (4538/8840), (OR 0.65, (95% CI 0.52, 0.82) *p* = 0.0002), as shown in Figure 22, respectively.

## 4. Discussion

Liver transplantation (LT) represents the ideal treatment option for patients with HCC since it achieves radical tumor clearance and eradicates the underlying liver diseases. However, several patients on the waiting list for LT are faced with tumor progression, the loss of chance for transplantation, or even death due to severe organ shortage and long waiting list times [54]. Thus, in order to overcome the gap between the numbers of donors and recipients, salvage liver transplantation has been proposed in the last decade as an attractive and feasible strategy that combines liver resection and subsequent LT in the case of HCC recurrence [55,56,57,58,59].

This meta-analysis includes the highest number of articles comparing the findings of primary and salvage liver transplantation for HCC and also demonstrates completely new results compared to other studies on the same topic, bringing different and innovative concepts to the strategy of salvage transplantation for recurrence of HCC after liver resection [19].

Operating time and intraoperative blood loss are some of the surgical variables in terms of safety and feasibility most taken into consideration when Salvage LT is compared with Primary LT. Several studies reported a longer mean operating time for SLT than PLT. Although considerable differences exist in terms of duration of surgery and blood loss among the included articles, the duration of the operation and the extent of bleeding are necessarily affected by some technical and anatomical issues. SLT increases the difficulty of surgery due to severe adhesion in the abdominal operation area and due to abnormal anatomical structures as a consequence of previous hepatic resection [22,60]. Our meta-analysis revealed a significantly longer duration of surgery for SLT than Primary LT and also disclosed that intraoperative bleeding was slightly higher in the Salvage LT strategy, but this finding was not statistically significant. Moreover, differences between the two surgical approaches in terms of the mean need for intraoperative RBC and FFP transfusion were not statistically significant. Several papers showed that innovations in surgical techniques and accumulation of surgical experience have gradually decreased the risk of perioperative bleeding for SLT. It has been shown that reducing intraoperative bleeding and blood transfusions rate leads to a better postoperative recovery.

Our study showed that the reoperation rate was significantly higher in the SLT group than in the PLT group, and the perioperative mortality was slightly higher for the Salvage LT approach. Multiple preoperative bridging and downstaging treatments and the liver resection before salvage LT led to the formation of dense adhesions, portal collateral circulations due to hypertension, and coagulopathy [61,62,63]. Therefore, these factors increase bleeding after SLT, likely accounting for the higher re-exploration rate. Surgery for salvage liver transplantation is technically demanding, and this could explain the slightly higher perioperative mortality in patients undergoing Salvage rather than Primary LT.

In terms of intensive care unit stay and length of hospital stay, our data showed a longer recovery in the SLT group, although these findings were not statistically significant. On the other hand, overall vascular complication rate, overall infection, and sepsis rate were statistically similar between the two groups.

Several studies have found that the outcome of patients with HCC was similar between liver resection and liver transplantation [15,64,65,66]. Therefore, liver resection and LT are not opposing alternatives, but, rather, represent the components of a combined strategy for the management of HCC: liver resection can potentially improve the survival of patients listed for LT by decreasing the risk of dropout [67]. Moreover, minimally invasive liver resection (MILR) has a minor technical impact on a subsequent liver transplantation and seems to be associated with shorter operation time, reduced blood loss, and transfusion requirement during Salvage LT [68,69]. Therefore, MILR (laparoscopic or robotic) may become the gold standard for “early” HCC in patient cirrhosis and mild portal hypertension. In 2008, Felli et al. [70] introduced the concept of liver resection as a selection tool for LT. In fact, some pathological characteristics of the resected specimen can identify a subgroup of patients with favorable histological factors (small and well-differentiated HCC, without satellite nodules or microvascular invasion) who could avoid upfront LT because the risk of recurrence appears to be relatively low and if it should occur, then transplantation remains a salvage option at a later date [42,71,72,73]. On the other hand, patients showing negative prognostic histological features on the resected specimen (e.g., microvascular infiltration, high grade of differentiation) could undergo liver transplantation prior to tumor recurrence: so-called “de principe” SLT [17,74]. Indeed, the French allocation system recently integrated the SLT strategy within its algorithm, although no priority is given to patients at a high risk of HCC recurrence. These results and future research would clarify the role of the molecular and biological pattern of HCC in order to stratify patients with a high risk of recurrence and then arrive at defining the best personalized treatment [75].

A clear definition of “transplantability criteria in SLT”, that is, criteria that identify the group of patients who benefit most from transplantation for HCC recurrence after liver resection, has not yet been established [46,76]. Most authors agree that the criteria of patients with a limited recurrence within the Milan criteria is acceptable in order to achieve a good survival post-SLT [77]. Recently, Liu et al. observed the efficacy of SLT for patients with recurrent HCC after liver resection within the University of California San Francisco (UCSF) criteria, since in that study there was no significant difference in OS and DFS rates between the SLT and PLT groups [41].

Recurrence of HCC after transplantation is still a devastating event as no surgical or pharmacological therapy has shown significant prolongation of these patients’ survival [78,79,80,81]. Some authors have observed that the strategy of the salvage liver transplantation may increase the risk of recurrence of post-transplant patients, thus limiting their survival [51].

In our meta-analysis, the HCC recurrence rate was 15.4% in the SLT group and 10.9% in the PLT group. However, between the different studies taken into account by our meta-analysis, contrasting results can be observed with regard to tumor recurrence. Adam et al. [29]. reported that SLT had an increased risk of recurrence and poorer survival compared with primary transplantation. By contrast, in the same period, Belghiti et al. [30] showed that recurrence rate, operative mortality, and long-term survival were comparable between the two groups.

Important end points of this meta-analysis were overall survival (OS) rate and disease free-survival rate between SLT and PLT. Our meta-analysis showed statistically significant lower 1-, 3-, and 5-year DFS rates for SLT compared to PLT. However, DFS as a long-term outcome indicator could be misleading because it is a composite end point influenced by two events: death and tumor recurrence. However, to better determine long-term outcomes, future studies should match patients based on histological features (tumor size and nodule number) at explant pathology which clearly influence tumor recurrence and mortality [72].

The 1-year OS rate presented no significant difference between SLT and PLT, whereas 3- and 5-year overall survival rates were significantly slightly lower in SLT than after PLT. However, previous studies disclosed that the 5-year survival rates did not differ significantly for patients with SLT and for those with PLT (69% vs 73%; *p* = 0.34) [39].

Our results on survival post-SLT appear to be in contrast to the recent meta-analyses published on the subject [19,20,21]. However, this is not surprising because most of the studies included in the meta-analysis show a lower survival in the SLT group than in the PLT groups [39]. On the other hand, while survival differences often do not reach a statistically significant difference within an individual study, this difference in survival becomes statistically significant in the meta-analysis, which represents a statistical tool of great relevance and precision (since it “weights” the result in individual studies according to its precision) [82].

Despite the relatively high quality of the included articles, there are several limitations concerning this meta-analysis. The included studies were retrospective and not randomized, so the variables analyzed exhibited heterogeneity. However, the heterogeneity within the studies was treated and resolved by applying the random effect model on all the variables in the study [83]. Moreover, some studies included heterogeneous patient populations with transplantation for HCC recurrence and those who underwent SLT due to liver failure, although this latter indication represents less than 5% of the SLT. Therefore, because of the inherent risk of bias in the considered articles, it is desirable that further well-designed studies are conducted.

Nevertheless, our systematic review summarizes most of the available evidence in comparing outcomes of SLT and PLT. To our knowledge, it is the largest and most recent meta-analysis that makes these comparisons. It introduces completely new results that can form the scientific basis on which to develop further studies on the topic of liver transplantation as an integrated therapy in the treatment of HCC.

## 5. Conclusions

Our meta-analysis advocates the relative safety and feasibility of SLT over the PLT approach for patients with HCC. Specifically, the results of our study confirm that SLT offers comparable technical outcomes but slightly lower survival outcomes with respect to PLT.

## Figures and Tables

**Figure 1 cancers-14-03465-f001:**
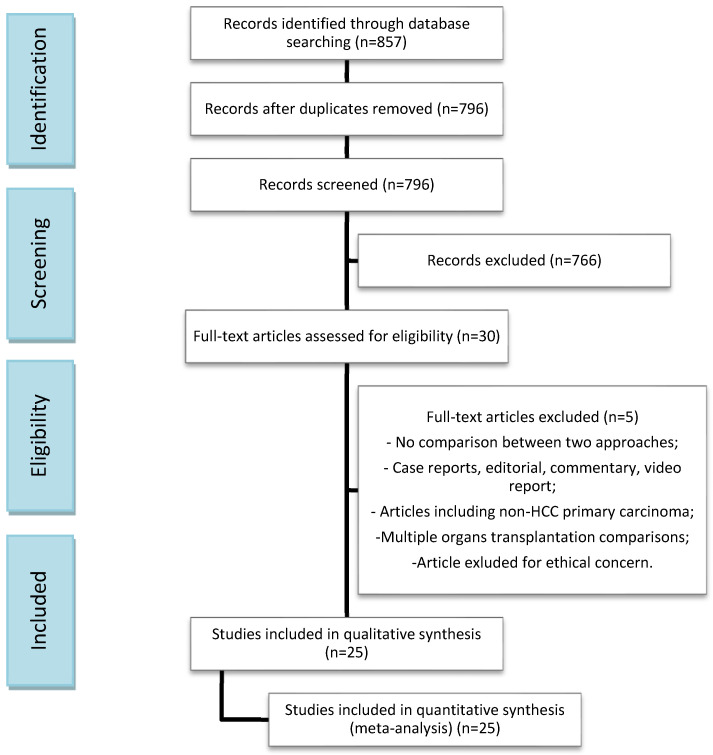
Search flow diagram.

**Figure 2 cancers-14-03465-f002:**
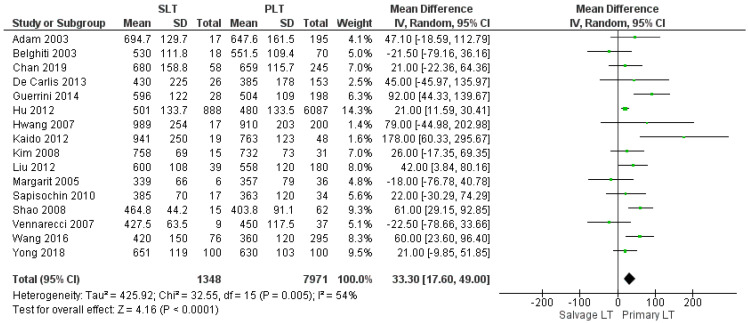
Operating time.

**Figure 3 cancers-14-03465-f003:**
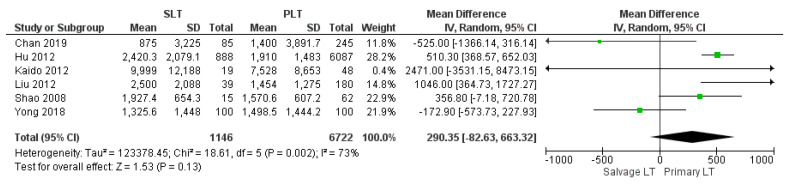
Intraoperative blood loss.

**Figure 4 cancers-14-03465-f004:**
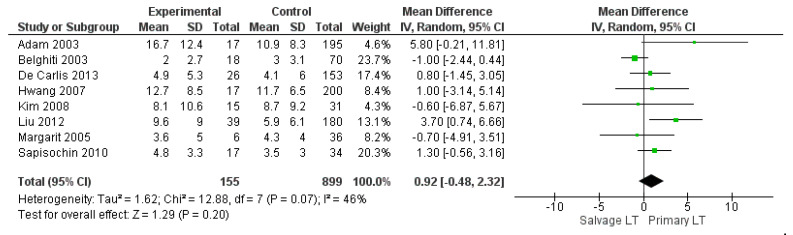
Intraoperative Red Blood Cell (RBC) transfusion.

**Figure 5 cancers-14-03465-f005:**
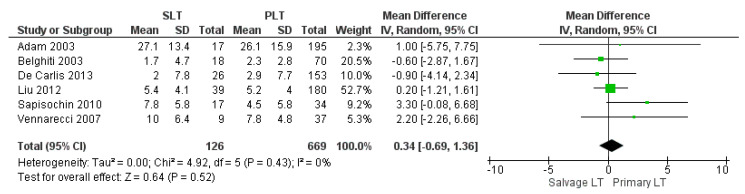
Fresh frozen plasma (FFP) transfusion.

**Figure 6 cancers-14-03465-f006:**
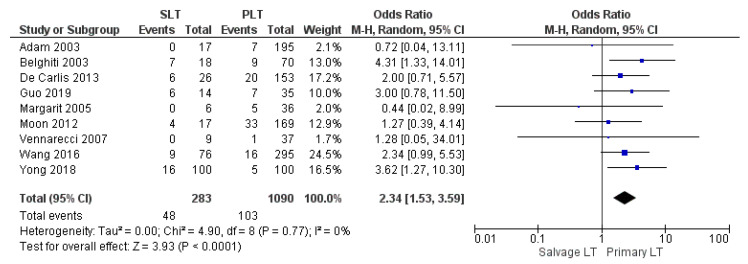
Reoperation rate.

**Figure 7 cancers-14-03465-f007:**
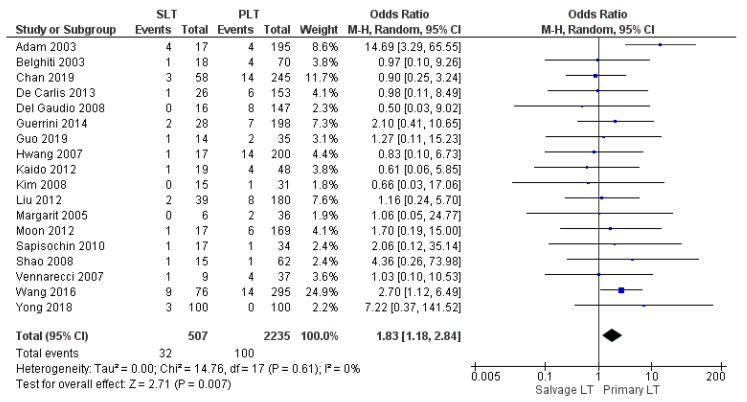
Perioperative mortality rate.

**Figure 8 cancers-14-03465-f008:**
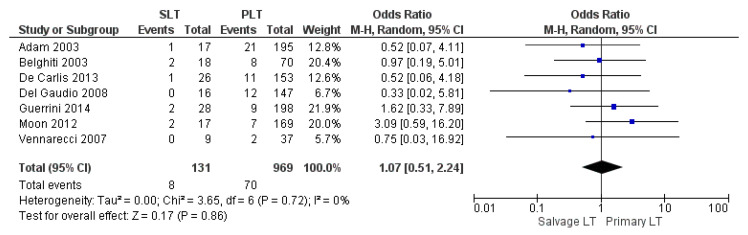
Retransplantation rate.

**Figure 9 cancers-14-03465-f009:**
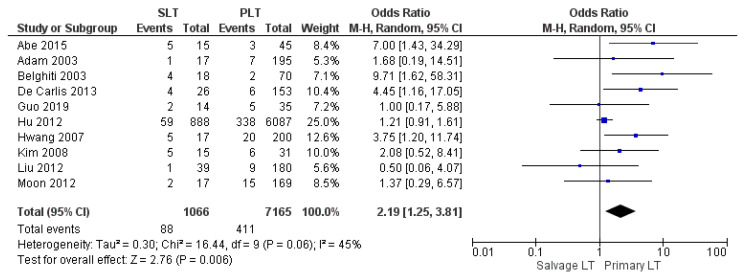
Postoperative bleeding.

**Figure 10 cancers-14-03465-f010:**
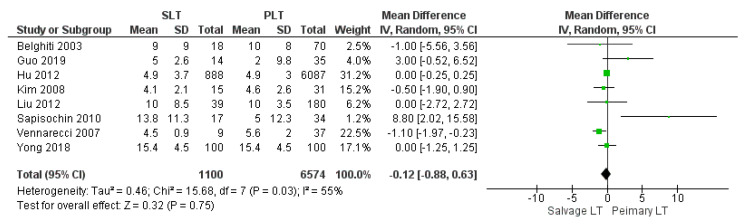
Intensive care unit stay.

**Figure 11 cancers-14-03465-f011:**
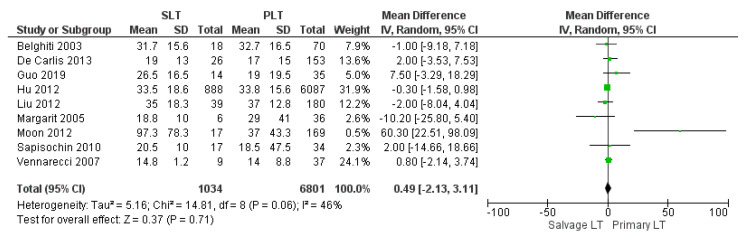
Length of hospital stay.

**Figure 12 cancers-14-03465-f012:**
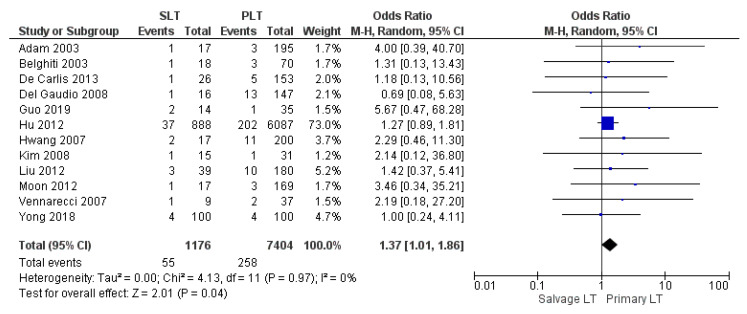
Overall vascular complication.

**Figure 13 cancers-14-03465-f013:**
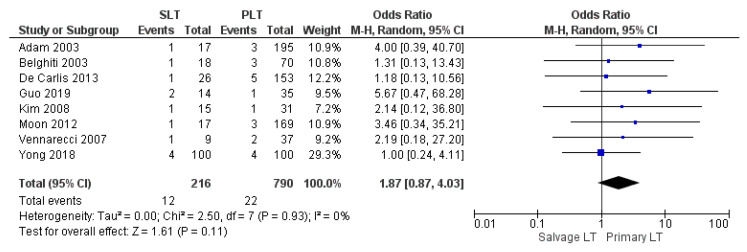
Arterial thrombosis.

**Figure 14 cancers-14-03465-f014:**
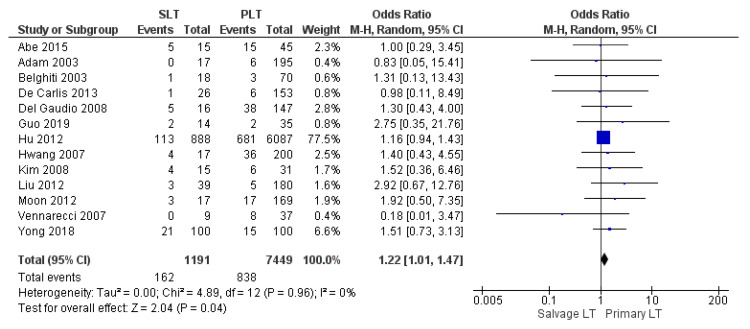
Biliary complication.

**Figure 15 cancers-14-03465-f015:**
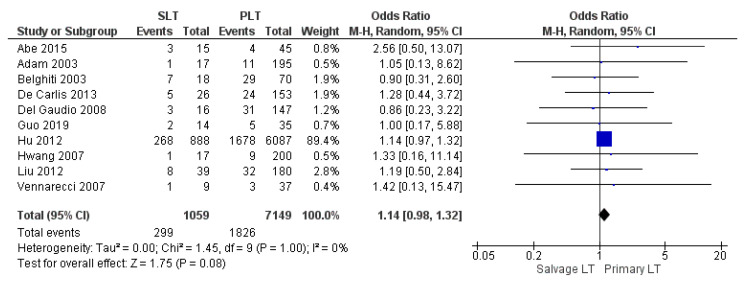
Infection and sepsis.

**Figure 16 cancers-14-03465-f016:**
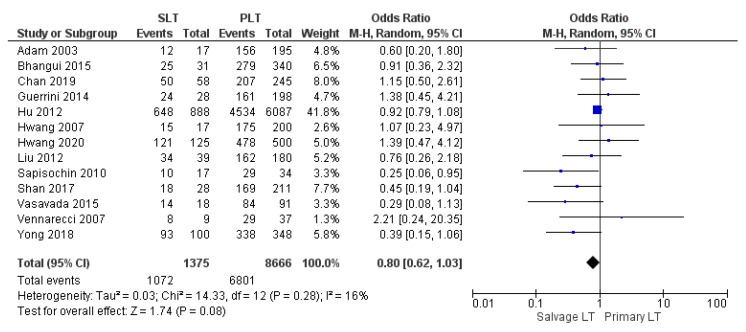
1-year overall survival rates.

**Figure 17 cancers-14-03465-f017:**
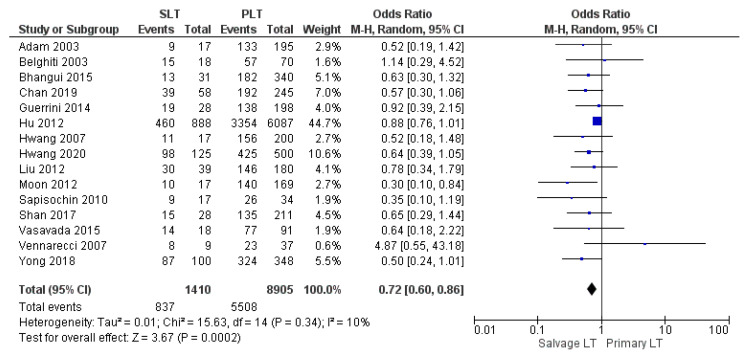
3-year overall survival rates.

**Figure 18 cancers-14-03465-f018:**
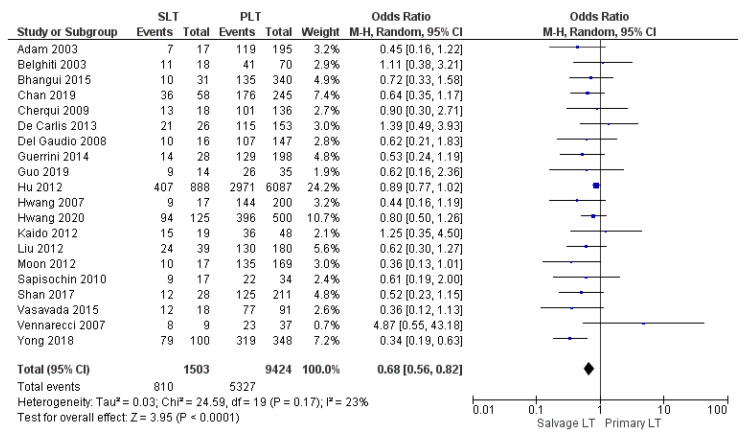
5-year overall survival rates.

**Figure 19 cancers-14-03465-f019:**
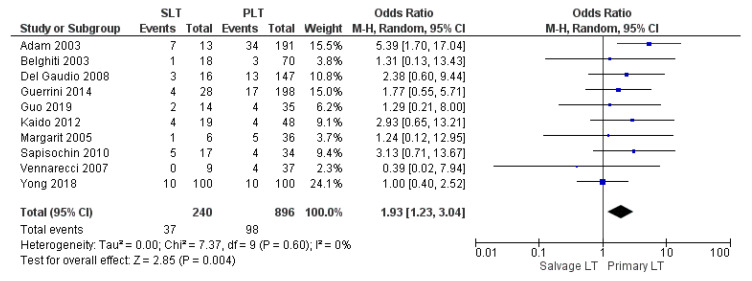
HCC recurrence rate.

**Figure 20 cancers-14-03465-f020:**
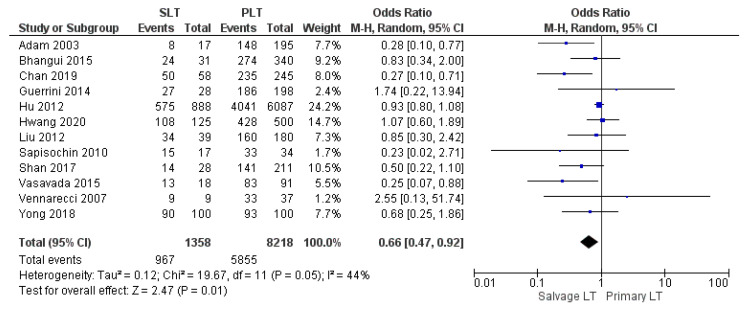
1-year disease free survival rates.

**Figure 21 cancers-14-03465-f021:**
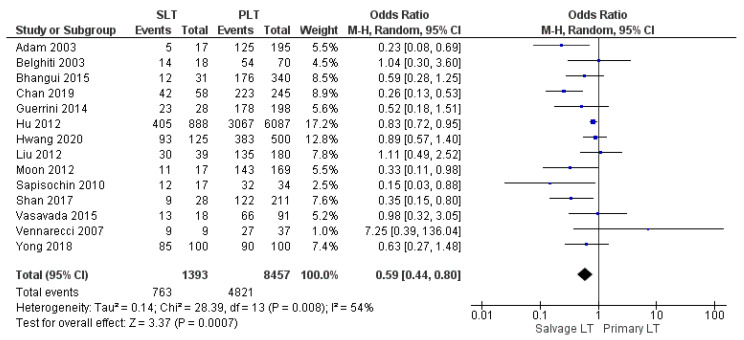
3-year disease free survival rates.

**Figure 22 cancers-14-03465-f022:**
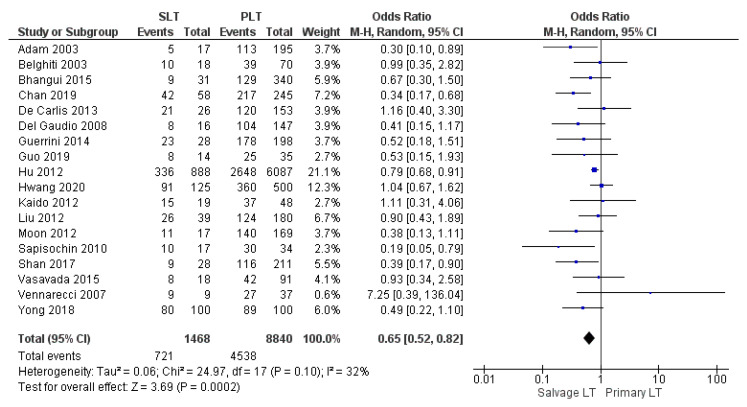
5-year disease free survival rates.

**Table 1 cancers-14-03465-t001:** Summary of studies included in the Meta-analysis.

n.	Author	Region	Year	Study Period	Study Design	Sample Size		Follow-Up (mo)		LDLT/DDLT	MINORS (Quality)
						SLT	PLT	SLT	PLT		
1	Adam [29]	France	2003	1984–2000	OCS (R)	17	195	49	51	DDLT	21
2	Belghiti [30]	France	2003	1991–2001	OCS (R)	18	70	56.2	56.2	DDLT	21
3	Margarit [31]	Spain	2005	1988–2002	OCS (P)	6	36	NA	NA	NA	20
4	Hwang [32]	Korea	2007	1997–2006	OCS (R)	17	200	30.7	40.1	LDLT	22
5	Vennarecci [33]	Italy	2007	2001–2006	OCS (P)	9	37	26.3	26.3	NA	23
6	Del Gadio [34]	Italy	2008	1996–2005	OCS (R)	16	147	26.2	36	DDLT	23
7	Kim [35]	Korea	2008	2005–2007	OCS (NA)	15	31	18.3	18.7	DDLT + LDLT	20
8	Shao [36]	China	2008	2003–2005	OCS (P)	15	62	18	22.4	DDLT	22
9	Cherqui [37]	France	2009	1990–2007	OCS (R)	18	136	57.6	576	DDLT	21
10	Sapisochin [38]	Spain	2010	1990–2007	OCS (P)	17	34	70	70	NA	22
11	Hu [39]	China	2012	1999–2009	OCS (R)	888	6087	15.2	15	DDLT + LDLT	22
12	Kaido [40]	Japan	2012	1999–2009	OCS (R)	19	48	77	77	LDLT	22
13	Liu [41]	China	2012	2001–2011	OCS (R)	39	180	30	33	DDLT + LDLT	22
14	Moon [42]	Korea	2012	1996–2008	OCS (R)	17	169	27.3	39	LDLT	21
15	De Carlis [43]	Italy	2013	2000–2009	OCS (R)	26	153	NA	NA	NA	22
16	Guerrini [44]	Italy	2014	2000–2011	OCS (P)	28	198	44.2	44.2	DDLT + LDLT	22
17	Abe [45]	Japan	2015	2001–2011	OCS (R)	15	45	66.3	73.2	LDLT	22
18	Bhangui [46]	France	2015	1990–2012	OCS (P)	31	340	62	62	DDLT	23
19	Vasavada [47]	China	2015	2002–2012	OCS (R)	18	91	NA	NA	LDLT	22
20	Whang [48]	China	2016	2001–2011	OCS (P)	76	295	32.4	32.4	DDLT	23
21	Shan [49]	China	2017	2006–2015	OCS (R)	28	211	35	35	DDLT + LDLT	21
22	Yong [50]	Taiwan	2018	2000–2015	OCS (R)	100	100	NA	NA	LDLT	22
23	Chan [51]	Taiwan	2019	2001–2018	OCS (R)	58	245	NA	NA	LDLT	22
24	Guo [52]	Singapore	2019	2006–2017	OCS (P)	14	35	43.9	43.9	DDLT + LDLT	22
25	Hwan [53]	Korea	2020	2007–2018	OCS (R)	125	500	NA	NA	LDLT	23

**Table 2 cancers-14-03465-t002:** General and Patients characteristics.

	SLT	PLT	Patient (Studies)
**Total patients included**	**1630**	**9645**	**11,275 (25)**
Follow-up (months)	41.3	43.8	19
HBV infection (%)	1166/1399 (83.3)	7157/8652 (82.7)	16
HCV infection (%)	103/1240 (8.3)	786/7842 (10)	10
MELD score	11	14	12
AFP (ng/dl) pre-LT	184.2	208.4	11
MILAN in pre-Lt (%)	264/419 (63)	1683/2391 (70.4)	15
MILAN IN on explant (%)	183/268 (68.2)	702/948 (74)	4
Pre-LT Locoregional Treatments (%)	812/1221 (66.5)	2901/7600 (38.2)	11
Waiting list time (months)	9.6	7.2	6
Maximum tumor diameter pre LT (cm)	2.6	2.6	4
Maximum tumor diameter on explant (cm)	2.6	2.9	12
Number of HCC nodule pre LT	2	1.6	4
Number of HCC nodule on explant	3.3	2	9
Sum of tumor size on explant (cm)	3.1	3.8	4
Microvascular invasion (%)	145/491 (29.5)	394/1860 (21.2)	13

**Table 3 cancers-14-03465-t003:** Technical and postoperative outcomes; Oncological and survival outcomes.

Technical and postoperative outcomes				
Surgical outcome	Type of surgery	Observations (n)	Mean or %	Studies included (n)
Operating time (min)	SLT	1348	600.44	16
	PLT	7971	547.12	
Blood loss (ml)	SLT	1146	3174.55	6
	PLT	6722	2342.02	
RBC transfusion	SLT	155	7.8	8
	PLT	899	6.5	
FFP transfusion	SLT	126	9	6
	PLT	669	8	
Reoperation rate	SLT	48/283	16.9%	9
	PLT	103/1090	9.4%	
Mortality rate	SLT	32/507	6.3%	18
	PLT	100/2235	4.5%	
Re-transplantation rate	SLT	8/131	6.1%	7
	PLT	70/969	7.2%	
Postoperative bleeding	SLT	88/1066	8.25%	10
	PLT	411/7165	5.73%	
ICU stay (days)	SLT	1100	8.34	8
	PLT	6574	5.44	
Hospital stay (days)	SLT	1034	33.01	9
	PLT	6801	26.44	
Vascular complication	SLT	55/1176	4.68%	12
	PLT	258/7404	3.48%	
Arterial thrombosis	SLT	12/216	5.56%	8
	PLT	22/790	2.78%	
Biliary complication	SLT	162/1191	13.6%	13
	PLT	838/7449	11.2%	
Infection and sepsis	SLT	299/1059	28.2%	10
	PLT	1826/7149	25.5%	
Oncological and survival outcomes				
Oncological outcome	Type of surgery	Observations (n)	%	Studies included (n)
1-yr OS	SLT	1072/1375	77.9%	13
	PLT	6801/8666	78.5%	
3-yr OS	SLT	837/1410	59.3%	15
	PLT	5508/8950	61.9%	
5-yr OS	SLT	810/1503	53.9%	20
	PLT	5327/9424	56.5%	
HCC recurrence	SLT	37/240	15.4%	10
	PLT	98/896	10.9%	
1-yr DFS	SLT	967/1358	71.2%	12
	PLT	5855/8218	71.2%	
3-yr DFS	SLT	763/1393	54.8%	14
	PLT	4821/8457	57%	
5-yr DFS	SLT	721/1468	49.1%	18
	PLT	4538/8840	51.3%	

## Data Availability

Not applicable.

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
