# Peer review of "Salvage versus Primary Liver Transplantation for Hepatocellular Carcinoma: A Twenty-Year Experience Meta-Analysis"

_cancers, 2022, doi:10.3390/cancers14143465_

Round 1

Reviewer 1 Report

Gian Piero and colleagues performed Meta-analysis to compare salvage liver transplant (SLT) vs. primary liver transplant (PLT) for HCC in order to achieve some consensus about feasibility and safety of SLT from the oncology aspect as well as perioperative aspect. 

Although this manuscript is overall well written, I have a few concerns. The concept is not necessarily new as there was a very similar meta-analysis in 2018 (Yadav DK, Ann Transplant, 2018;23:524-45). Sixteen out of 25 studies analyzed in the current article were used in the previous meta-analysis. Hence, authors should emphasize more why they performed this study now and what is the new findings compared to the previous studies, instead of simply adding a few new studies. 

Next, in Discussion, authors stated that HCC recurrence is more often in SLT compared to PLT. Authors cited two conflicting papers here, without discussing why HCC recurrence is higher in the SLT group in the current study. Authors should add more detailed explanation as to why HCC recurrence was found more often in the SLT group. 

Authors touched upon future prospective trial as to the exact definition of "SLT criteria", which is understandable for formal definition. However, this Reviewer would expect slightly more in-depth discussion here as to what kind of criteria authors have in their mind etc. 

In Limitation, authors mentioned that some studies included did not distinguish the indication for transplant between HCC recurrence vs. advanced liver failure. This heterogeneity seems to be very critical to this reader; hence, authors should clarify this even further. 

Here are also some minor comments.

1) Authors should explain de principe SLT in the Introduction for the readers who are not familiar with the concept. 

2) some grammatical and typos throughout the manuscript. For example, in Line 53, 5-year DSF in Introduction (should have been DFS). Also, Line 281, although these finds (should have been findings) etc. 

Author Response

Comments and Suggestions for Authors

Gian Piero and colleagues performed Meta-analysis to compare salvage liver transplant (SLT) vs. primary liver transplant (PLT) for HCC in order to achieve some consensus about feasibility and safety of SLT from the oncology aspect as well as perioperative aspect. 

Although this manuscript is overall well written, I have a few concerns. The concept is not necessarily new as there was a very similar meta-analysis in 2018 (Yadav DK, Ann Transplant, 2018;23:524-45). Sixteen out of 25 studies analyzed in the current article were used in the previous meta-analysis. Hence, authors should emphasize more why they performed this study now and what is the new findings compared to the previous studies, instead of simply adding a few new studies. 

Next, in Discussion, authors stated that HCC recurrence is more often in SLT compared to PLT. Authors cited two conflicting papers here, without discussing why HCC recurrence is higher in the SLT group in the current study. Authors should add more detailed explanation as to why HCC recurrence was found more often in the SLT group. 

Authors touched upon future prospective trial as to the exact definition of "SLT criteria", which is understandable for formal definition. However, this Reviewer would expect slightly more in-depth discussion here as to what kind of criteria authors have in their mind etc. 

In Limitation, authors mentioned that some studies included did not distinguish the indication for transplant between HCC recurrence vs. advanced liver failure. This heterogeneity seems to be very critical to this reader; hence, authors should clarify this even further. 

I thank you very much for your careful and pertinent suggestions that I have taken into account in changing large parts in the discussion section (the changes are written in red). I think the article is now much more accurate and interesting for the readers of this important Journal.

I reply to your observations:
1) “The concept is not necessarily new as there was a very similar meta-analysis in 2018 (Yadav DK, Ann Transplant, 2018;23:524-45). Sixteen out of 25 studies analyzed in the current article were used in the previous meta-analysis. Hence, authors should emphasize more why they performed this study now and what is the new findings compared to the previous studies, instead of simply adding a few new studies. “

I made clear in several parts of the discussion that our meta-analysis introduces new and different concepts with respect to the meta-analysis of YADAD, emphasizing that our study is not a simple UpToDate meta-analysis, but shows new results compared to the studies already published on the subject.

2) Next, in Discussion, authors stated that HCC recurrence is more often in SLT compared to PLT. Authors cited two conflicting papers here, without discussing why HCC recurrence is higher in the SLT group in the current study. Authors should add more detailed explanation as to why HCC recurrence was found more often in the SLT group. 

This point has been expanded further in the discussion section trying to make it clear that unfortunately the HCC recurrence in the SLT is higher than the PLT group. This result explains why DFS survival is lower in the salvage transplant group.

3) Authors touched upon future prospective trial as to the exact definition of "SLT criteria", which is understandable for formal definition. However, this Reviewer would expect slightly more in-depth discussion here as to what kind of criteria authors have in their mind etc. 

I have explained what is meant by SLT Criteria and I argued the discussion by trying to explain which patients have a greater benefit from SLT.

4) In Limitation, authors mentioned that some studies included did not distinguish the indication for transplant between HCC recurrence vs. advanced liver failure. This heterogeneity seems to be very critical to this reader; hence, authors should clarify this even further. 

We have reviewed this variable in all individual meta-analysis studies. The indication to SLT for liver failure after liver resection is about 5%. Therefore it does not constitute a bias so relevant to the interpretation of the results.

Here are also some minor comments.

  • Authors should explain de principe SLT in the Introduction for the readers who are not familiar with the concept. 

We have clarified this concept in the introduction, as you have suggested.

  • some grammatical and typos throughout the manuscript. For example, in Line 53, 5-year DSF in Introduction (should have been DFS). Also, Line 281, although these finds (should have been findings) etc. 

              We have modified these grammatical errors, as you suggested.

Reviewer 2 Report

First, I want to congratulate the authors on this well written paper. Indeed, in the last few decades it’s becoming more and more difficult providing quality data on such a vast topic as transplant treatment options in HCC. Making a study such as this one will help to improve our understanding of the complicated relationship between pre-transplant treatments and post-transplant HCC outcome and progression. 

Since this is a meta-analysis and not a single center data I understand if some of the queries are beyond your explanation. I also would like to make some comments to potentially improve this paper:

-        It will be nice to have a reference on the statement on pg2 line 50-52

-        The salvage liver transplantation strategy was conceived for initially resectable and transplantable HCC to obviate upfront transplantation, with salvage LT in the case of recurrence. In your data are all livers from the SLT group post liver resection? In Tab 2 there is reported Pre – LT treatments which for the SLT group is 66.5% - is this liver resection or other interventional radiology treatments?

-        One of the papers cited in your manuscript by Prashant Bhangui et al  described drop out in the SLT group of 66%, similar drop out is described in other papers, what is unclear is what criteria have been used in order to select which patient will receive SLT or will be dropped out. Do you have any information how this selection has been done and maybe put these patient’s characteristics in a chart.

-        In 2018 Dipesh Kumar Yadav et al. conducted similar study (“Salvage Liver Transplant versus Primary Liver Transplant for Patients with Hepatocellular Carcinoma”) where SLT was compared with PLT for HCC. They found that SLT had superior 1-year, 3-year, and 5-year OS and DFS compared with that of PLT. How can you comment that your findings different so much from this study?

Author Response

First, I want to congratulate the authors on this well written paper. Indeed, in the last few decades it’s becoming more and more difficult providing quality data on such a vast topic as transplant treatment options in HCC. Making a study such as this one will help to improve our understanding of the complicated relationship between pre-transplant treatments and post-transplant HCC outcome and progression. 

 I thank you very much and I am happy that our article is of interest to you. I have taken all your comments into consideration, modifying some parts of the discussion section. Here are the answers to your comments:

Since this is a meta-analysis and not a single center data I understand if some of the queries are beyond your explanation. I also would like to make some comments to potentially improve this paper:

-        It will be nice to have a reference on the statement on pg2 line 50-52

       I provided the reference as you suggested.

-        The salvage liver transplantation strategy was conceived for initially resectable and transplantable HCC to obviate upfront transplantation, with salvage LT in the case of recurrence. In your data are all livers from the SLT group post liver resection? In Tab 2 there is reported Pre – LT treatments which for the SLT group is 66.5% - is this liver resection or other interventional radiology treatments?

        The table, to which you refer, reported the loco-regional treatments carried out before liver transplantation (Tace and RF) in order to reduce the burden of disease (downstaging) or prevent tumor progression (bridging); there are no reported cases of resection in the SLT group.

       Regarding the other part of your question, over 95% of cases of SLT are carried out for HCC after liver resection while 5% of SLT for liver failure post-resection. This has been checked in every single study and is explained in the discussion section.

-        One of the papers cited in your manuscript by Prashant Bhangui et al  described drop out in the SLT group of 66%, similar drop out is described in other papers, what is unclear is what criteria have been used in order to select which patient will receive SLT or will be dropped out. Do you have any information how this selection has been done and maybe put these patient’s characteristics in a chart.

       This observation is very important. Dropout of patients with HCC listed for liver transplantation in literature studies generally ranges from 5-10%. Certainly, of patients who have been resected for HCC, only 40-50% of cases then undergo LT for their HCC recurrence: either because the tumor recurrence is outside the criteria of Transplantability or because they are too old. The very interesting study that you mentioned is very articulate, because it also takes into account the intention to treat analysis from the time of diagnosis of HCC: that’s why the dropout rates in this study appears high. Therefore in the discussion section I have clarified what is meant and what are the Criteria of Transplantability in SLT.

-        In 2018 Dipesh Kumar Yadav et al. conducted similar study (“Salvage Liver Transplant versus Primary Liver Transplant for Patients with Hepatocellular Carcinoma”) where SLT was compared with PLT for HCC. They found that SLT had superior 1-year, 3-year, and 5-year OS and DFS compared with that of PLT. How can you comment that your findings different so much from this study?

              This observation is very important. Therefore, I made clear in several parts of the discussion that our meta-analysis introduces new and different concepts with respect to the meta-analysis of YADAD, emphasizing that our study is not a simple UpToDate meta-analysis, but shows new results compared to the studies already published on the subject. The survival rates post-LT are always lower in the SLT vs PLT group. These differences do not reach statistical significance in the individual studies, while our meta-analysis instead clearly shows that unfortunately overall and DFS are worse in the SLT group. This data can be inferred from the fact that the rate of tumor recurrence is higher in the SLT group so the DFS is worse and ultimately survival is affected by the death of these patients due to the recurrence.

Round 2

Reviewer 1 Report

significantly improved and responded to this reviewer's inquiries appropriately.